# Low Temperature Conditioning Reduced the Chilling Injury by Regulating Expression of the Dehydrin Genes in Postharvest Huangguan Pear (*Pyrus bretschneideri* Rehd. cv. Huangguan)

Yudou Cheng [1,2], Jingang He [1,2] 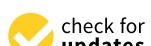, Yunxiao Feng [1,2], Jiangli Zhao [1,2] and Junfeng Guan [1,2,*]

1 Institute of Biotechnology and Food Sciences, Hebei Academy of Agriculture and Forestry Sciences, Shijiazhuang 050051, China
2 Plant Genetic Engineering Center of Hebei Province, Shijiazhuang 050051, China
* Correspondence: junfeng-guan@263.net; Tel.: +86-31-1876-52118

**Abstract:** 'Huangguan' pear (*Pyrus bretschneideri* Rehd. cv. Huangguan) fruit is sensitive to chilling injury (CI), which exhibits peel browning spots (PBS) during cold storage. Dehydrin (DHN) is considered to be related to cold tolerance in plants, but its function in postharvest pear fruit during storage remains unclear. In this study, six *PbDHNs* (*PbDHN1–6*) genes were identified and characterized, and the PbDHN proteins were sorted into $Y_nK_n$, $SK_n$ and $Y_nSK_n$ according to the major conserved motifs related to the number and location of K-segments, S-segments, and Y-segments. In addition, there were five cold-responsive related *cis*-acting elements in the promoter region of the *PbDHNs*. The analysis of fruit quality suggested that compared with a storage temperature at 20 °C, a storage temperature of 0 °C results in CI in 'Huangguan' pear fruit, while a storage temperature of 10 °C and low temperature conditioning (LTC) alleviates the CI. Moreover, gene expression results indicated that the six *PbDHNs* were markedly enhanced at low temperatures, especially at 0 °C. The transcripts of *PbDHN1*, *PbDHN4*, *PbDHN5* and *PbDHN6* were also increased in the fruit stored at 10 °C, but they were lower than that at 0 °C except *PbDHN5*. Compared with low temperature storage at 0 °C, LTC treatment significantly depressed the expression of *PbDHN1*, *PbDHN2*, *PbDHN3*, *PbDHN4*, and *PbDHN6*, while enhanced the mRNA amount of *PbDHN5*. In conclusion, *PbDHN1*, *PbDHN4*, *PbDHN5*, and *PbDHN6* were closely related to the CI, and LTC lowered the CI by down-regulating the expression of *PbDHN1*, *PbDHN4*, and *PbDHN6* and by up-regulating *PbDHN5* in 'Huangguan' pear fruit.

**Keywords:** low temperature conditioning; pear; dehydrins; chilling injury

## 1. Introduction

'Huangguan' (*Pyrus bretschneideri* Rehd. cv. Huangguan) is one of the most important cultivars of pear in China, and it is favored by consumers because of the regular shape and excellent quality. However, 'Huangguan' pear fruit is prone to PBS, which is a CI phenomenon that develops during cold storage [1–3], which results in the loss of fruit commodity value.

Dehydrin (DHN) is a class of hydrophilic, thermostable stress-responsive proteins that is widely present in plants [4]. DHN is characterized depending on the presence of three major conserved motifs (K-, Y-, and S-segments). The K-segment has a conserved lysine-rich 15-amino acid motif (EKKGIMDKIKEKLPG), which is usually located at the C-terminal region of DHN. The known DHNs contain at least one K-segment. These K-segments can form a putative amphiphilic α-helix structure that can interact with both membranes and denatured proteins, maintaining the stability of the membranes to protect cell function when the plant has suffered from multiple abiotic stresses, such as osmotic, drought, salinity and low temperature stresses [5–12]. Many DHNs also contain one or more Y-segments (DEYGNP) located at the N-terminus; these segments display amino

acid identity at the nucleotide binding site motif of chaperone proteins. The S-segment contains multiple serine residues that can be modified through phosphorylation, which may be involved in ion-binding [13]. According to the number and distribution of the three conserved motifs, the DHNs are divided into five subclasses: $K_n$, $Y_nK_n$, $K_nS$, $SK_n$, and $Y_nSK_n$ [14].

The function of DHNs involved in the response to cold stress has been widely studied [3,12–20]. In *Arabidopsis*, the transcripts of the four characterized *DHN* genes of *AtRAB18* (Responsive to ABA 18), *AtCOR47* (Cold-regulated 47), *AtLTI29* (Low temperature-induced 29), and *AtLTI30* (Low temperature-induced 30) accumulate in response to low temperature. Moreover, co-overexpressing *AtRAB18* and *AtCOR47* or co-overexpressing *AtLTI29* and *AtLTI30* could significantly improve the cold tolerance of the *Arabidopsis* seedling [21,22]. Twelve *DHN* genes (*MdDHNs*) were identified in apple (*Malus domestica*), and four *DHNs* in leaves, *MdDHN2*, *MdDHN4*, *MdDHN6*, and *MdDHN8*, were found to be strongly induced by cold treatment (4 °C) [23]. The grapevine (*Vitis vinifera*) *DHN* genes family comprises four divergent members (*VvDHN1–4*), although only *VvDHN1* and *VvDHN2* appeared to respond to cold treatment [24]. Freezing treatment (−3 °C) resulted in the transcripts of all the seven *EjDHNs* (*EjDHN1–7*) increasing in loquat (*Eriobotrya japonica*) fruit [25]. In *citrus unshiu*, the *DHN* gene *CuCOR19* (Cold-regulated 19) was found to be cold-responsive; furthermore, over-expression of *CuCOR19* could prevent lipid peroxidation and subsequently enhance cold tolerance in transgenic tobacco [26]. However, the function of *DHNs* on the CI development of the postharvest fruit is still little known.

In this study, to clarify the members of *DHNs* family and their role in the CI development in 'Huangguan' pear fruit under low temperature storage, six *DHN* genes were identified and their expression patterns subsequently analyzed under different storage temperatures.

## 2. Materials and Methods

### 2.1. Materials and Treatments

'Huangguan' pear fruit with paper-bagging were harvested at commercial maturity (average single fruit weight: 317.20 ± 35.45 g; soluble solid content: 12.16 ± 0.19%; starch content: 1.88 g/kg; firmness: 68.22 ± 3.50 N) from an orchard located in Jinzhou City (115 04′34.80″ E, 38 03′13.50″ N), Hebei Province, and then immediately transported to the laboratory within 2 h. After being removed from the paper bag, the pear fruit with uniform weight and shape without visible defects were selected as materials, and the treatments were carried out as follows: The fruit were divided into four groups. The first, second and third groups were directly stored at 20 ± 1 °C (20 °C), 10 ± 0.5 °C (10 °C), and 0 ± 0.5 °C (0 °C), respectively, while the fourth group was first stored at 10 ± 0.5 °C for 3 days and then transferred to 0 ± 0.5 °C and marked as low temperature conditioning (LTC).

For all treatments, the CI was observed at 0, 3, 5, 10, 15, 20, and 30 days of storage, and then the peel samples were quickly frozen in liquid nitrogen and stored at −80 °C until used. Three replicates were performed for each treatment with 10 fruit per replicate.

### 2.2. Cloning of PbDHN Genes

Total RNA was extracted from the peel sample by the improved CTAB method [27]. After digestion by DNAse-I, the first strand cDNA was synthesized using the AMV reverse transcription kit (TaKaRa Biomedicals, Dalian, China) according to the manufacturer's instruction. Primers for cloning of *DHNs* were designed based on the genomic sequences of *Pyrus bretschneideri* (https://www.ncbi.nlm.nih.gov/, accessed on 15 September 2019) using the Omega 2.0 software. The primer sequences are listed in Table 1.

**Table 1.** Sequence of primers for the *PbDNHs* cloning.

| Gene | Forward Primer | Reverse Primer |
|---|---|---|
| *PbDHN1* | 5′-TCAGTTTGAAACAATGGCGC-3′ | 5′-GTTCACACAGTCACGCCAAG-3′ |
| *PbDHN2* | 5′-AGAAAATGGCGCATTTACAAAATC-3′ | 5′-GCTCACACCTAGAGGCACCAG-3′ |
| *PbDHN3* | 5′-TATCCAAACTTCCAGACTCAAATTG-3′ | 5′-ACCGTTACTCGAACGAGCAC-3′ |
| *PbDHN4* | 5′-TTTGAAATATGGCGAATTATGGT-3′ | 5′-AGACGATGATCTACTTCTTGTGGC-3′ |
| *PbDHN5* | 5′-CAACATTTCATTTGCTCTCTTCAC-3′ | 5′-TTCTTTCCGAACACAAGACAAGA-3′ |
| *PbDHN6* | 5′-CAGTTTCATTTGTTTGTTTATCTTTTG-3′ | 5′-ACCGAGGTCAGACCAGACAC-3′ |

### 2.3. Multiple Sequence Alignment, Gene Structure Construction, and Phylogenetic Analysis

The exon-intron structures of the *PbDHNs* were determined from the alignments of cDNA and pear (*Pyrus bretschneideri*) genomic sequences using the software GSDS 2.0 (http://gsds.cbi.pku.edu.cn, accessed on 25 October 2019). The protein molecular weight and isoelectric point of the six isolated PbDHN proteins were predicted using the Prot-Param program (http://au.expasy.org/tools/protparam. html, accessed on 23 October 2019) based on their amino acid compositions. The deduced amino acid sequences were aligned using the software Clustal X. The DHN proteins sequences of arabidopsis (*Arabidopsis thaliana*) were obtained from the TAIR website (http://www.arabidopsis.org, accessed on 6 November 2019): AtCOR47 (At1g20400), AtLTI29 (At1g20450), AtHIRD11 (At1g54410), AtERD14 (At1g76180), AtLTI30 (At3g50970), AtXERO1 (At3g50980), AtRAB18 (At5g66400), At2G21490, At4G38410, and At4G39130. DHNs sequences of peach (*Prunus persica*) were downloaded from the GDR database (http://www.rosaceae.org, accessed on 19 November 2019): PpDHN1 (ppa005514m), PpDHN2 (ppa011637m), PpDHN3 (ppa010326m), PpDHN4 (ppa026861m), PpDHN5 (ppa010975m), and PpDHN6 (ppa009997m). DHNs sequences of apple (*Malus domestica*), loquat (*Eriobotrya japonica*), grape (*Vitus yeshanensis*), Chinese plum (*Prunus mume*), sorghum (*Sorghum bicolor*), pepper (*Capsicum annuum*), and tomato (*Solanum lycopersicum*) were downloaded from the NCBI database (http://www.ncbi.nlm.nih.gov, accessed on 20 November 2019). Apple: MdDHN1 (JQ649456), MdDHN2 (JQ649457), MdDHN3 (JQ649458), MdDHN4 (JQ649459), MdDHN5 (JQ649460), MdDHN6 (JQ649461), MdDHN7 (JQ649462), MdDHN8 (JQ649463), and MdDHN9 (JQ649464). Loquat: EjDHN1 (FJ472835), EjDHN2 (FJ472836), EjDHN3 (KF277187), EjDHN4 (KF277188), EjDHN5 (KF277189), EjDHN6 (KF277190), and EjDHN7 (KF277191). Grape: VyDHN1 (JF900497), VyDHN2 (JQ408442), VyDHN3 (JQ408443), and VyDHN4 (JQ408444). Chinese plum: PmLEA8 (XM_016796210.1), PmLEA10 (XM_008243512.2), PmLEA19 (XM_008237879.1), PmLEA20 (XM_008238721.1), and PmLEA29 (XM_008222942.1). Sorghum: SbDHN1 (KT865881) and SbDHN2 (KT780443). Pepper: CaDHN1 (JX402924). Tomato: SlDHN (KF585024.1). Based on the result of multiple alignment, a phylogenetic tree was generated using the MEGA 5.0 software (http://www.megasoftware.net/mega.html, accessed on 11 December 2019) using a neighbor-joining method with 1000 bootstrap replicates.

### 2.4. Prediction of Cis-Acting Elements in Promoter Region

The promoter sequences representing approximately 1500 bp upstream of the translation start codon of *PbDHNs* were obtained from the NCBI database (https://www.ncbi.nlm.nih.gov/, accessed on 10 January 2020), and the putative *cis*-acting elements in the upstream acting regions of the *PbDHNs* were scanned using the PlantCARE database (http://bioinformatics.psb.ugent.be/webtools/plantcare/html/, accessed on 12 January 2020).

### 2.5. Fruit Quality and Estimation of CI

The fruit firmness was measured using the GY-4 digital fruit hardness meter (TOP Instruments Co., Hangzhou, Zhejiang, China) at two equidistant points on the equatorial region with the skin removed. The soluble solids content (SSC) was measured using a PAL-1 pocket digital refractometer (ATAGO Co., Ltd., Tokyo, Japan). Titratable acidity (TA)

was determined using a base buret by titration with 0.01 mol/L NaOH up to pH 8.1, and the TA content was expressed as malic acid (%).

The CI index was evaluated according to the PBS score that was described by a previous report [3]: 0 (no peel browning spots), 1 (area of the peel browning spots < 25%), 2 (area of the peel browning spots $\geq$ 25% but < 50%), and 3 (area of the peel browning spots $\geq$ 50%). The CI index was calculated using the following formula:

$$\text{CI index} = \Sigma \text{ (peel browning spots scale} \times \text{number of fruit at the scale level)}/[3 \times \text{(total number of fruit)}]$$

### 2.6. Quantitative RT-PCR (qRT-PCR) Analysis

The First-strand cDNA was synthesized using the PrimeScript$^{RT}$ Reagent Kit with gDNA Eraser (TaKaRa Biomedicals, Dalian, China) according to the manufacturer's recommended protocol. The reactions contained 10.0 µL SYBR Green PCR Premix Ex Taq™ (TaKaRa Biomedicals, Dalian, China), 0.5 µL of ROX Reference DyeII, 0.5 µL of forward and reverse primer (Each for 10 µmol/L), 5 ng of cDNA, and 6.0 µL of ddH$_2$O. The qRT-PCR was performed using the ABI 7500 instrument (Applied Biosystems, Foster City, CA, USA), and the cycling parameters were as follows: 10 s at 95 °C, 40 cycles of 95 °C for 5 s, and 60 °C for 34 s. The primers of *PbDHN* genes for qRT-PCR were designed using the software of OMIGA 2.0, and they are listed in Table 2. The *PbActin2* (*PbACT2*) was used to normalize the amount of gene-specific qRT-PCR products.

**Table 2.** Primers of *PbDHN*s for qRT-PCR.

| Gene | Forward Primer | Reverse Primer |
|------|---------------|----------------|
| *PbDHN1* | 5′-TACTCATCTCATACGACCTCCA-3′ | 5′-GGTGTCCACCGGGAAGTT-3′ |
| *PbDHN2* | 5′-ACTTGGCCATCACGGTGC-3′ | 5′-CTGCTGTGGTGGCAGCAT-3′ |
| *PbDHN3* | 5′-TACTACAGGTGCCACCACCG-3′ | 5′-GAGCACACCAGTGACACCAT-3′ |
| *PbDHN4* | 5′-TCAACACAGTCGGACTGATA-3′ | 5′-CCATGACGTCCAACAATCAC-3′ |
| *PbDHN5* | 5′-GAAGGGTATGACGGACAAGA-3′ | 5′-GGTCATCCCTATTCCCACCT-3′ |
| *PbDHN6* | 5′-AAGCTGCCAGGTGGGAAT-3′ | 5′-TCTCCTGTGCGCCCTGT |
| *PbACT2* | 5′-GGACATTCAACCCCTCGTCT-3′ | 5′-ATCCTTCTGACCCATACCAACC-3′ |

### 2.7. Statistical Analysis

The software GraphPad Prism 4 was used to prepare the figures. All values were expressed as mean $\pm$ SD. The results were statistically analyzed with ANOVA using SPSS 18 software (SPSS Inc., Chicago, IL, USA). Differences among treatments were assessed through Duncan's multiple comparison test ($p < 0.05$).

## 3. Results

### 3.1. Identification and Classification of DHN Genes in 'Huangguan' Pear Fruit

Based on the RNA-Seq data of 'Huangguan' pear fruit (Data unpublished), six candidates *PbDHNs* with FPKM > 1 were selected for the present study. After being confirmed by RT-PCR cloning, sequencing, and aligned with the NCBI genome database, six *PbDHNs* with the length of CDS varied from 564 to 1245 bp were identified. The sequences of *PbDHN1*, *PbDHN4*, and *PbDHN6* were completely consistent with three cold-shock protein CS120-like genes (XM_009347774.2, XM_009347773.1, and XM_009347771.2, respectively), *PbDHN2* was identical with a dehydrin DHN2-like (XM_009347776.2), *PbDHN3* was identical with a dehydrin DHN3-like (XM_00934778.1), and *PbDHN5* was identical with *COR47* (XM_009336070.1). In addition, the six *PbDHNs* were mapped to 2 of 17 pear chromosomes, and *PbDHN1*, *PbDHN2*, *PbDHN3*, *PbDHN4*, and *PbDHN6* were located on Chr 2, while *PbDHN5* was present on Chr 15. To verify whether these genes were dehydrins, the protein sequences of the six *PbDHNs* were aligned online (www.expasy.org/tools/, accessed on 23 October 2019) to predict the basic parameters, and all of the six PbDHNs contained the PF00257 (Dehydrin) conserved domain. The analysis of protein characteristic revealed

that six deduced PbDHNs varied from 187 (PbDHN3) to 474 (PbDHN4) in amino acids length, the predicted molecular weights were between 19.77 kDa (PbDHN3) and 50.33 kDa (PbDHN4), and the isoelectric points ranged from 5.25 (PbDHN5) to 7.98 (PbDHN3) (Table 3).

**Table 3.** Characteristics analysis of *PbDHNs*.

| Gene | Access No | Length of CDS/bp | Number of Amino Acid | Pfam | Molecular Weight/kDa | Isoelectric Points | Chromosome Localization |
|---|---|---|---|---|---|---|---|
| *PbDHN1* | XM_009347774.2 | 603 | 200 | PF00257 | 21.27 | 6.86 | chr2: 6074631−6074029 |
| *PbDHN2* | XM_009347776.2 | 651 | 216 | PF00257 | 22.88 | 7.38 | chr2: 6054467−6052893 |
| *PbDHN3* | XM_009347778.1 | 564 | 187 | PF00257 | 19.77 | 7.98 | chr2: 6057707−6056344 |
| *PbDHN4* | XM_009347773.1 | 1245 | 474 | PF00257 | 50.33 | 7.43 | chr2: 6071938−6070037 |
| *PbDHN5* | XM_009336070.1 | 861 | 286 | PF00257 | 33.08 | 5.25 | chr15: 116768−118339 |
| *PbDHN6* | XM_009347771.2 | 586 | 194 | PF00257 | 20.26 | 6.92 | chr2: 6062552−6061487 |

### 3.2. Multiple Sequence Alignment, Structure, and Phylogenetic Analysis of PbDHNs

The alignment was performed using Clustal X, indicating that the amino acid sequence of each dehydrin was quite different except for the domain Y-, K-, and S-segments. PbDHN2, PbDHN3, PbDHN5, and PbDHN6 contained three K-segments, PbDHN1 contained four K-segments, and PbDHN4 contained nine K-segments. With the exception of PbDHN1 that had no S-segment, the other five PbDHNs contained one S-segment. PbDHN1, PbDHN2, PbDHN4, and PbDHN6 included one Y-segment, while PbDHN3 had two Y-segments, and PbDHN5 did not contain Y-segments (Figure 1).

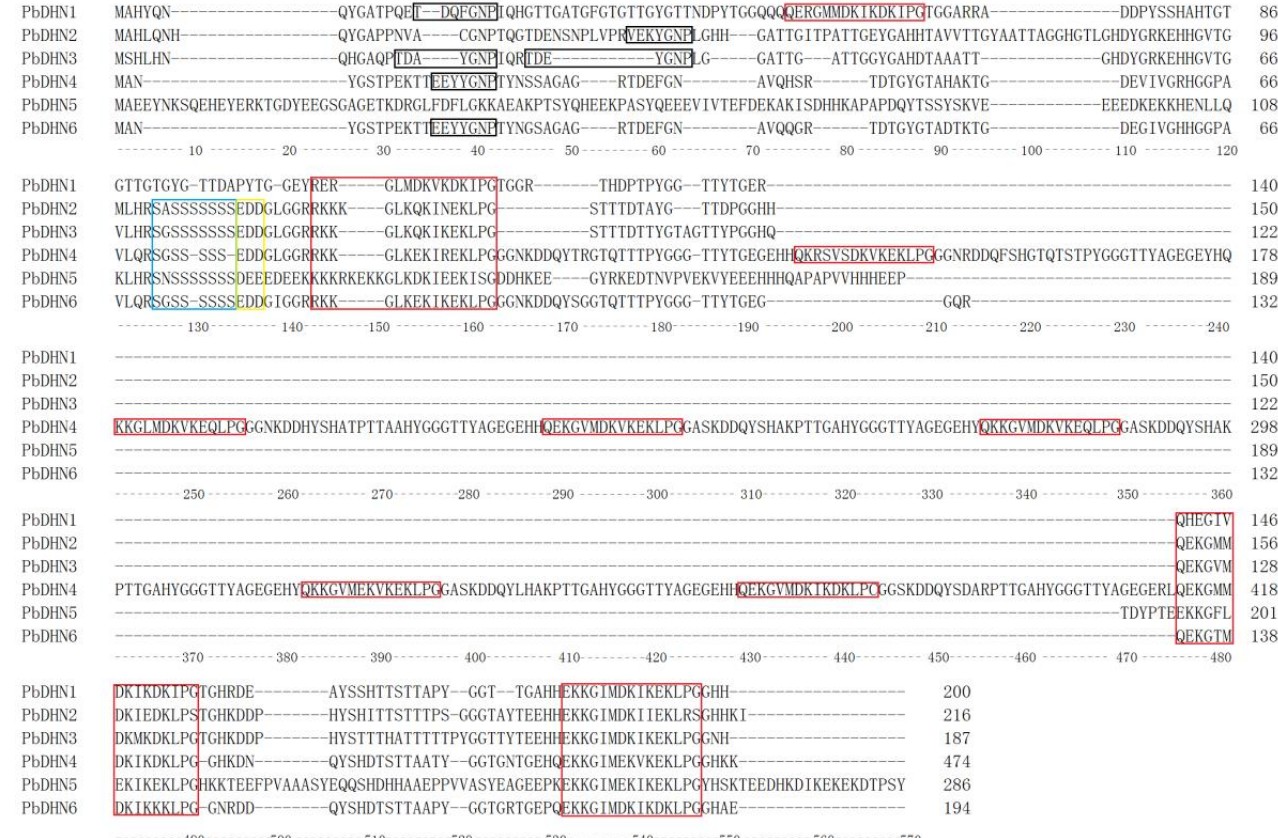

**Figure 1.** Alignment of multiple amino acid sequences of PbDHNs in 'Huangguan' pear fruit. The sequences in black boxes indicate Y-segment, the sequences in the blue box indicate S-segment, the sequences in red boxes indicate K-segment, and the sequences in the yellow box indicate phosphorylation site.

The six PbDHNs were sorted into three classes by neighbor-joining analysis, including class I ($Y_nSK_n$, containing PbDHN2, PbDHN3, PbDHN4, and PbDHN6), class II ($Y_nK_n$, containing PbDHN1), and class III ($SK_n$, containing PbDHN5) (Figure 2A). The intron-exon structure analysis was determined by aligning the cDNA to pear genomic sequences, and it was found that *PbDHN1* contained only one exon, while the other five *PbDHNs* contained two exons and one intron (Figure 2B).

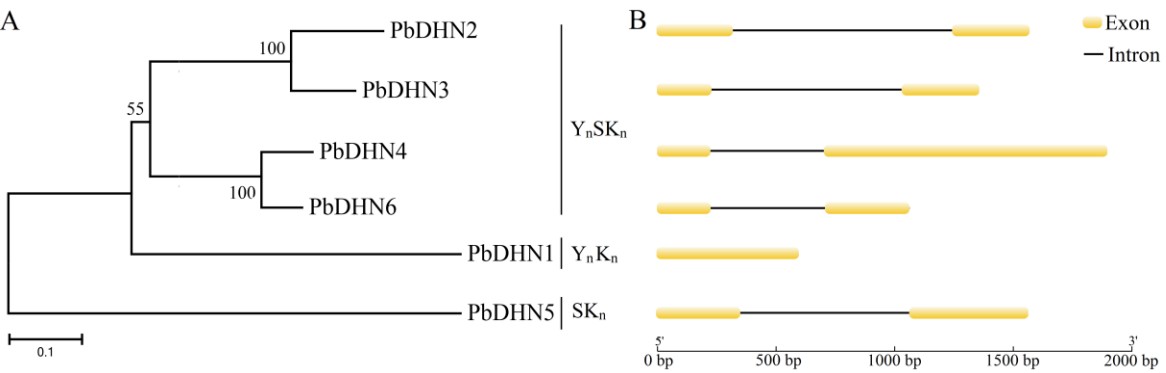

**Figure 2.** Phylogenetic relationships (**A**) and gene structure (**B**). The phylogenetic tree of PbDHN proteins was performed using MEGA 5.0 software with the neighbor-joining method. The bootstrap test was performed with 1000 replicates. Percentage bootstrap scores of >50% were displayed. Exons were indicated by yellow round-corner rectangles and introns by black lines.

To study the phylogenetic relationships among different plant species, an unrooted phylogenetic tree based on sequence alignments of the DHN protein sequences of pear (*Pyrus bretschneideri*), apple (*Malus domestica*), peach (*Prunus persica*), Chinese plum (*Prunus mume*), loquat (*Eriobotrya japonica*), grape (*Vitus yeshanensis*), arabidopsis (*Arabidopsis thaliana*), sorghum (*Sorghum bicolor*), pepper (*Capsicum annuum*), and tomato (*Solanum lycopersicum*) was constructed to identify putative orthologs via MEGA5.0. The results revealed that DHN proteins from different species could be roughly divided into four groups: I, II, III, and IV. Group I included PbDHN5, two MdDHNs (MdDHN8 and Md-DHN9), two EjDHNs (EjDHN2 and EjDHN5), two VyDHNs (VyDHN2 and VyDHN3), two PpDHNs (PpDHN3 and PpDHN4), two PmDHNs (PmLEA19 and PmLEA29), and three AtDHNs (AtCOR47, AtLTI29, and AtEDR14). Group II included two PbDHNs (PbDHN2 and PbDHN3), two MdDHNs (MdDHN1 and MdDHN7), one EjDHN (EjDHN1), one PpDHN (PpDHN2), and one PmDHN (PmLEA10). Group III included one PbDHN (Pb-DHN1), two MdDHNs (MdDHN5 and MdDHN6), two EjDHNs (EjDHN4 and EjDHN6), one PpDHN (PpDHN6), and one PmDHN (PmLEA8). Group IV included two PbDHNs (PbDHN4 and PbDHN6), three MdDHNs (MdDHN2, MdDHN3, and MdDHN4), two EjDHNs (EjDHN3 and EjDHN7), two VyDHNs (VyDHN1 and VyDHN4), two PpDHNs (PpDHN1 and PpDHN5), one PmDHN (PmLEA20), and two AtDHNs (AtRAB18 and AtXREO1) (Figure 3). The results of this phylogenetic analysis suggested that the six PbDHNs were closely related to similar such proteins of other plant species.

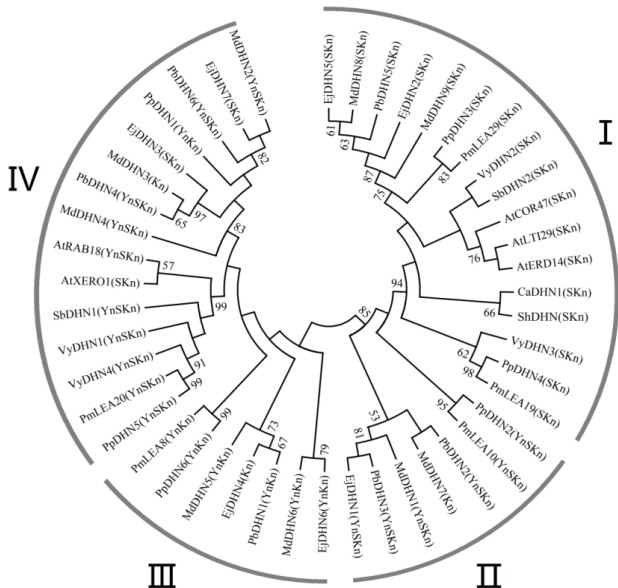

**Figure 3.** Phylogenetic tree analysis of DHNs from different plant species. The unrooted phylogenetic tree was constructed using MEGA 5.0 with the neighbor-joining method. The bootstrap test was performed with 1000 replicates. Percentage bootstrap scores of >50% were displayed. vThe results revealed that DHN proteins from different species could be roughly divided into four groups: I, II, III, and IV.

### 3.3. Comparison of Cis-Acting Elements in the Promoter Region of PbDHNs

To reveal the regulation mechanism of *PbDHNs*, the promoter region, including 1500 bp of sequence upstream of the start codon of six *PbDHNs*, were scanned for the putative *cis*-acting elements using the PlantCARE database. The six *PbDHNs* mainly contained four sorts of *cis*-acting elements related to cold stress, including a ABA response element (ABRE/ACGTG), a drought response element (DRE/GCCGAC), MYB (C/TAACCA), and MYC (CAA/TT/GTG), as well as a ethylene response element (ERE/ATTTCATA). There were obvious differences in the abundance and distribution of *cis*-acting elements among the promoter regions of the six *PbDHNs*. The promoter region of *PbDHN1* had six ABREs, three MYBs, and three MYCs. *PbDHN2* contained one ABRE and six MYBs. *PbDHN3* contained seven ABREs, one DRE, two EREs, three MYBs, and three MYCs. *PbDHN4* contained three ABREs, two DREs, three MYBs, and two MYCs. *PbDHN5* contained seven ABREs, two DREs, two EREs, two MYBs, and four MYCs, while *PbDHN6* contained five ABREs, two DREs, two EREs, one MYB, and four MYCs (Figure 4).

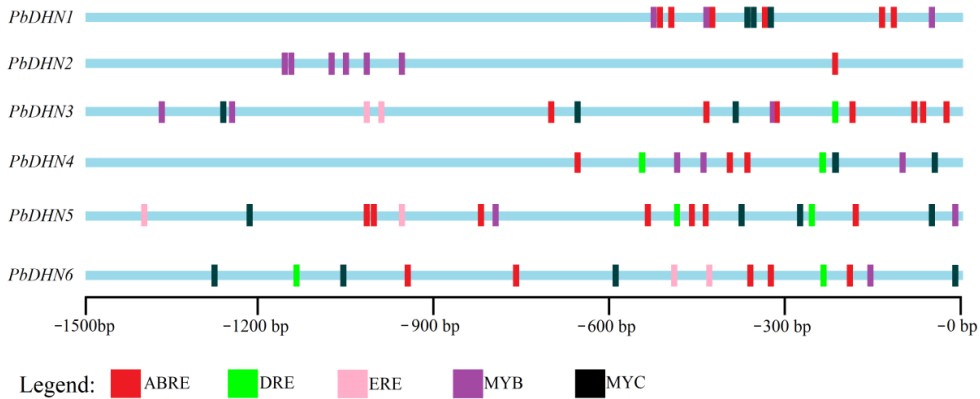

**Figure 4.** Location of putative *cis*-acting elements in the promoter regions of *PbDHNs*. The promoter sequences were obtained from 1500 bp upstream of the translation start codon (ATG), and the putative cold stress-related *cis*-acting elements were presented.

### 3.4. The fruit Quality and CI Index

The firmness, SSC, and TA content were the important factors of fruit quality. After 30 days of storage, the firmness and TA content of pear fruit stored at 20 °C were significantly lower than the fruit stored at 0 °C with LTC treatment, indicating the degree of more advanced ripening when stored at 20 °C. The TA content of pear fruit stored at 10 °C was also lower than the fruit stored at 0 °C with LTC treatment, while the firmness changed slightly, suggesting that the quality of the fruit stored at 10 °C began to decline. There was no obvious difference in firmness, SSC, and TA content between the fruit stored at 0 °C with LTC treatment (Table 4).

**Table 4.** Fruit quality of 'Huangguan' pear fruit with different treatments after 30 days of storage.

| Treatments | Firmness (N) | SSC (%) | TA (%) |
|---|---|---|---|
| 20 °C | 57.66 ± 3.77 [b] | 12.04 ± 0.39 [a] | 0.138 ± 0.002 [b] |
| 10 °C | 67.63±0.77 [a] | 12.53 ± 0.59 [a] | 0.144 ± 0.002 [b] |
| 0 °C | 66.09 ± 2.43 [a] | 12.00 ± 0.31 [a] | 0.165 ± 0.002 [a] |
| LTC | 66.13 ± 2.00 [a] | 12.34 ± 0.40 [a] | 0.158 ± 0.005 [a] |

Different letters in the table indicated significant differences ($p < 0.05$).

The CI indexes of 'Huangguan' pear fruit under different temperature treatments were quite different. Compared with fruit stored at 20 °C (no CI development), the fruit stored at 0 °C appeared CI at day 10, and then the CI index increased sharply, reaching the highest value (0.441) at day 30. Although the fruit stored under 10 °C developed CI at day 10, the CI index was much lower than the fruit stored under 0 °C, and the highest value of CI index was only was 0.053. Compared with the fruit stored under 0 °C, the LTC-treated fruit displayed markedly lower CI index after 10 days of storage, and the highest CI index value was 0.078 (Figure 5).

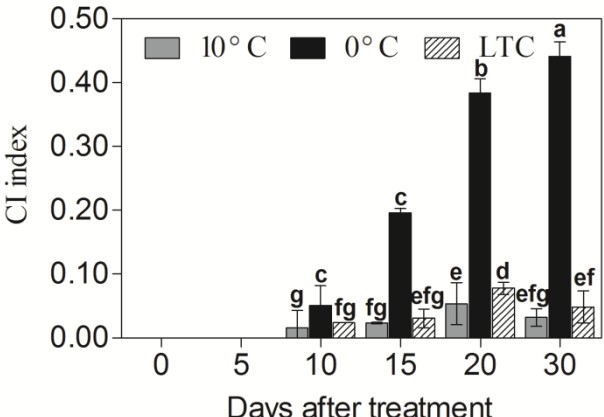

**Figure 5.** Estimation of CI in 'Huangguan' pear fruit at different temperature storage. Data are means of three replicates, and error bars indicate SD. Different letters (a–g) above the bar chart indicated significant differences ($p < 0.05$) by Duncan's multiple comparison test.

### 3.5. Expression Patterns of PbDHNs under Different Storage Temperatures

The expression profiles of the six *PbDHNs* were different when the fruit was stored at 0 °C, 10 °C, and 20 °C. When the fruit was stored at 20 °C, the expression profiles of *PbDHN1*, *PbDHN2*, and *PbDHN5* increased at first, and then decreased (Figure 6A,E,Q). The expression level of *PbDHN4* rapidly decreased when the fruit was stored at 20 °C, while the mRNA amounts of *PbDHN3* and *PbDHN6* changed slightly before day 20 of storage, and displayed higher expression level at day 30 (Figure 6I,M,U). When the fruit was stored at 10 °C, the expression levels of *PbDHN1*, *PbDHN2*, *PbDHN4*, and *PbDHN6* rapidly increased at day 3 of storage, and then decreased, while the expression levels *PbDHN3* and *PbDHN5* gradually increased till 20 days of storage (Figure 6B,F,J,N,R,V).

In addition, the mRNA amounts of *PbDHN1*, *PbDHN4*, *PbDHN5*, and *PbDHN6* in the fruit stored at 10 °C were much higher than the fruit stored at 20 °C. The expression levels of the six *PbDHNs* in the fruit stored at 0 °C rapidly increased at day 3 of storage, among which *PbDHN1*, *PbDHN2*, and *PbDHN4* reached the peak at day 15 of storage (Figure 6C,G,O), while *PbDHN3*, *PbDHN5*, and *PbDHN6* increased gradually until 30 days of storage (Figure 6K,S,W). Moreover, the mRNA amounts of *PbDHN1*, *PbDHN3*, *PbDHN4*, and *PbDHN6* in the fruit stored at 0 °C were much higher than the fruit stored at 20 °C and 10 °C. These results indicate that the six *PbDHNs* are low temperature-induced genes.

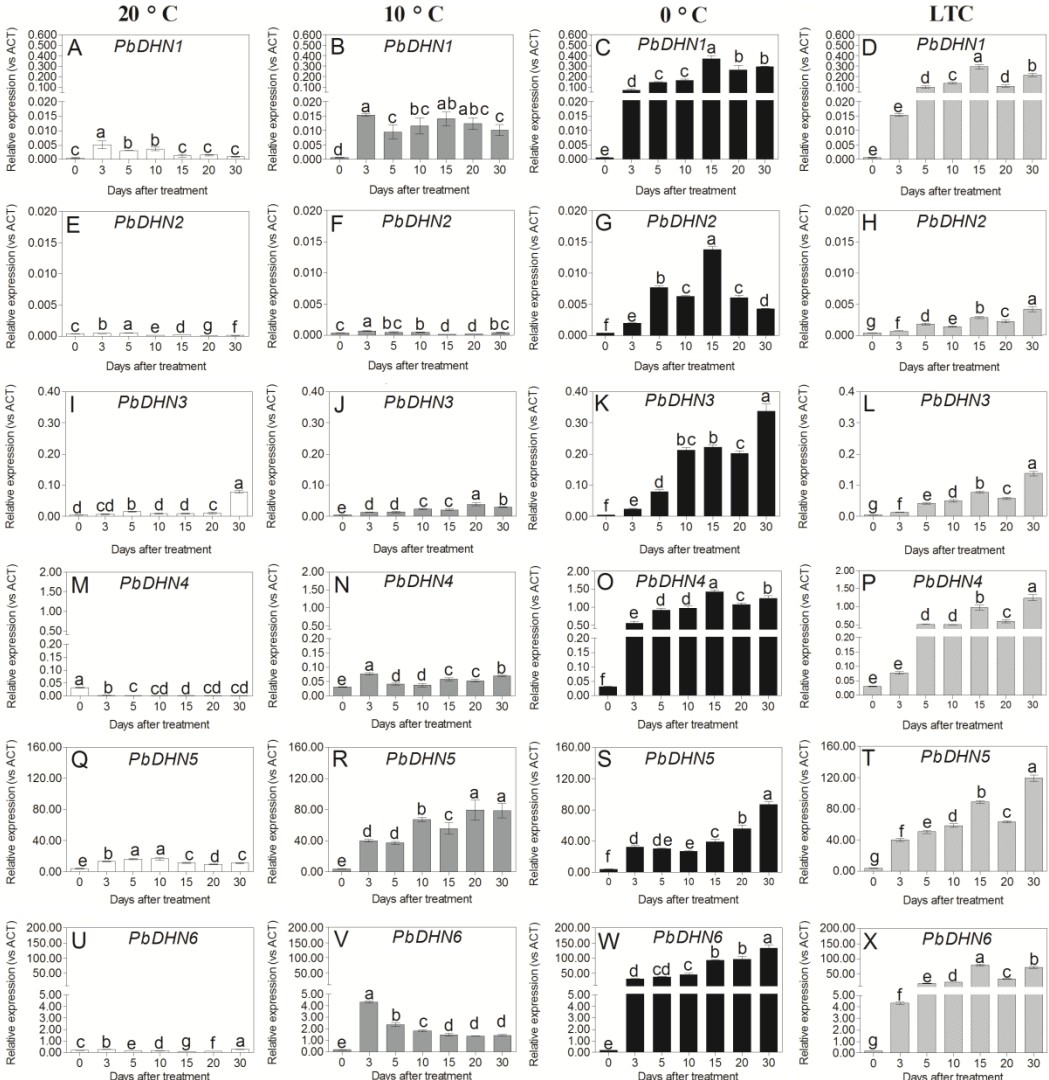

**Figure 6.** Effects of 20 °C (**A,E,I,M,Q,U**), 10°C (**B,F,J,N,R,V**), 0°C (**C,G,K,O,S,W**) storage and LTC (**D,H,L,P,T,X**) treatments on the expression pattern of *PbDHNs* in the peel of 'Huangguan' pear fruit. Data are means of three biological replicates, and error bars indicate SD. Different letters (a–g) above the bar chart indicated significant differences (*p* < 0.05) by Duncan's multiple comparison test.

### 3.6. Expression Patterns of PbDHNs under LTC Treatment

In the LTC-treated fruit, the expression levels of *PbDHN1* and *PbDHN6* markedly increased till 15 days of storage, and then decreased, while *PbDHN2*, *PbDHN3*, *PbDHN4* and *PbDHN5* increased until day 30 of storage. Compared with the fruit stored at 0 °C, the mRNA amount of *PbDHN1*, *PbDHN2*, *PbDHN3*, *PbDHN4* and *PbDHN6* were lower (Figure 6D,H,L,P,X), while *PbDHN5* was higher (Figure 6T).

## 4. Discussion

　　DHNs can protect proteins and membranes from unfavorable structural changes. The K-segment especially is believed to plan an important role in cold stress by interacting with various cell components [5,7,27–29]. The structure of two K-fragments connecting with φ fragments is believed to have the function of resisting cold stress [30]. The DHN with cold-responsive function should contain at least three K-fragments and the acidic and neutral type DHN ($K_n$, $SK_n$, and $Y_nK_n$) preferentially accumulate in response to cold stress [31]. In the present study, the six DHNs cloned from 'Huangguan' pear fruit were classified as $Y_nK_n$ (PbDHN1), $SK_n$ (PbDHN5), and $Y_nSK_n$ (PbDHN2, PbDHN3, PbDHN5, and PbDHN6) type, respectively, all of which contained at least three K-segments (Figures 1 and 2A), suggesting that they might be closely related to response to low temperature in 'Huangguan' pear fruit. Previous studies have confirmed that the *cis*-acting elements such as ABRE, DRE, MYB, and MYC were involved in the process of low temperature stress [4,32–37], and in this study the promoter regions of the six *PbDHNs* in 'Huangguan' pear fruit contained at least the ABRE and MYB *cis*-acting elements (Figure 4), so it further suggests that these *PbDHNs* were associated with the low temperature stress in 'Huangguan' pear fruit.

　　The expression profiles confirmed that the six *PbDHNs* of 'Huangguan' pear fruit were indeed induced by low temperature, among which *PbDHN1*, *PbDHN2*, *PbDHN3*, *PbDHN4*, and *PbDHN6* were up-regulated under 0 °C, while the *PbDHN5* was also induced by a moderate temperature treatment (10 °C) (Figure 6). It has been considered that the homologous genes of *PbDHNs* (Figure 3), such as the *EjDHN6*, *MdDHN6*, and *PpDHN6* (homolog to *PbDHN1*), the *EjDHN1* (homolog to *PbDHN2* and *PbDHN3*), the *VyDHN1*, *AtRAB18*, *MdDHN2*, and *MdDHN4* (homolog to *PbDHN4* and *PbDHN6*), as well as the *AtCOR47*, *AtLTI29*, *PmLEA19*, and *EjDHN5* (homolog to *PbDHN5*), were all involved in the cold stress response of plants [18,19,21,23–25,38]. Thus, it is suggested that the six *PbDHNs* participated in the cold response process of 'Huangguan' pear fruit.

　　The 'Huangguan' pear fruit easily suffers from CI when the fruit is directly stored at low temperature, while the LTC treatment significantly inhibits the occurrence of CI [1]. In this study, LTC treatment down-regulated the expression of the cold response genes such as *PbDHN1*, *PbDHN2*, *PbDHN3*, *PbDHN4*, and *PbDHN6* (Figure 6D,H,L,P,X), indicating that the LTC treatment could desensitize or delay the response to low temperature stress in fruit in the early stages of cold storage, and it could make fruit more suitable for enduring long-term cold stress. In addition, LTC-treated fruit displayed greater transcript of *PbDHN5* than the fruit at 0 °C storage, which was different from the other five *PbDHNs* (Figure 6T); therefore, the acting mechanism of *PbDHN5* was different [39]. It has been known that over-expression of *EjDHN5* and *SbDHN2* (homolog to *PbDHN5*) in tobacco can improve the ability to resist low temperature stress [40]. Furthermore, in *Arabidopsis thaliana*, knocking out the *PbDHN5* homologous gene *LTI29* can decrease tolerance to cold stress in mutant seedlings [41]. Thus, up-regulation of *PbDHN5* by LTC treatment might be beneficial in reducing CI in 'Huangguan' pear fruit during cold storage.

## 5. Conclusions

　　Six *PbDHNs* genes were cloned from 'Huangguan' pear fruit, and the six *Pb*DHN proteins belonged to three types: $Y_nK_n$, $SK_n$ and $Y_nSK_n$. Low temperature (0 °C) easily resulted in CI and markedly up-regulated the expression levels of the six *PbDHNs*. LTC treatment reduced the occurrence of CI in 'Huangguan' pear fruit, decreased the expression levels of *PbDHN1*, *PbDHN2*, *PbDHN3*, *PbDHN4*, and *PbDHN6*, while increasing the expression level of *PbDHN5*. Above all, the six *PbDHNs* were low temperature-responsive genes; as such, they might be involved in the occurrence of CI under low temperature storage. LTC could deduce the CI by regulating *PbDHN1*, *PbDHN4*, *PbDHN5*, and *PbDHN6* expression in 'Huangguan' pear fruit.

**Author Contributions:** Conceptualization, Y.C. and J.G.; methodology, Y.F.; software, J.H.; validation, Y.C., J.G., and J.Z.; formal analysis, J.Z.; investigation, Y.F.; resources, Y.C.; data curation, Y.C. and J.G.;

writing—original draft preparation, Y.C.; writing—review and editing, J.G.; project administration, Y.C. and J.G. All authors have read and agreed to the published version of the manuscript.

**Funding:** This research was funded by the Key R & D plan of Hebei Province (22327505D) and the HAAFS Agriculture Science and Technology Innovation (2022KJCXZX−SSS−3).

**Institutional Review Board Statement:** Not applicable.

**Informed Consent Statement:** Not applicable.

**Data Availability Statement:** The data presented in this study are available within the article.

**Conflicts of Interest:** The authors declare no conflict of interest.

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
