# Peer review of "Low Temperature Conditioning Reduced the Chilling Injury by Regulating Expression of the Dehydrin Genes in Postharvest Huangguan Pear (Pyrus bretschneideri Rehd. cv. Huangguan)"

_horticulturae, doi:10.3390/horticulturae8111022_

Round 1

Author Response

Dear  Reviewer:

Thank you very much for your comments and suggestions concerning our manuscript entitled “Low temperature conditioning reduced the chilling injury by regulating expression of the dehydrin genes in postharvest Huangguan pear (Pyrus bretschneideri Rehd. cv.Huangguan)” (Horticulturae-1967390). Those comments and suggestions are all valuable and very helpful for revising and improving our paper, as well as the important guiding significance to our researches. We have studied comments and suggestions carefully and agree with all of the changes in the manuscript. Revised portion are marked in red in the manuscript. The main corrections in the manuscript and the responds to the review comments are as flowing:

Point 1: Line 34: Peel browning is still a chilling injury. So why the authors say that that was considered?

Response 1: PBS is a postharvest physiological disorder in 'Huangguan' pear once the fruit are directly stored at a low temperature (0℃) without gradual cooling or precooling, including low temperature conditioning (LTC), while it rarely occurs in other pear varieties under directly cold storage. Our previous study has suggested that PBS is a chilling injury symptom, so we changed the sentence ‘which was considered as a chilling injury (CI) phenomenon during cold storage’ to ‘which is a chilling injury (CI) phenomenon during cold storage’. Please see the related references:

Li, D.; Cheng, Y.; Dong, Y.; Shang, Z.; Guan, J. Effects of low temperature conditioning on fruit quality and peel browning spot in ‘Huangguan’ pears during cold storage. Postharvest Biol. Technol. 2017, 13, 68–73.

Wei, C.; Ma, L.; Cheng, Y.; Guan, Y.; Guan, J. Exogenous ethylene alleviates chilling injury of ‘Huangguan’ pear by enhancing the proline content and antioxidant activity. Sci. Hortic. 2019, 257, 108671.

Point 2: Line 34: PBS is already defined in the abstract.

Response 2: Yes, we have changed.

Point 3: Line 42-51: The authors should develop more on what the names genes mean.

Response 3: Yes, we have done.

Point 4: Line-44-45: I think this information should appear ride after writing about DHN.

Response 4: Yes, we have done.

Point 5: Overall, I think the introduction is scarce and should be clearer

Response 5: Yes, we rephrased the introduction.

Point 6: Line 61: The first, the second and third group were treated for how long?

Response 6: In this study, we mainly pay attention to chilling injury development. Therefore, all treatments were observed within 30 days of storage. Under normal conditions, the ‘Huangguan’ pear can be stored for 25-30 days at room temperature (20-25°C), while more than 120 days at low temperature (0 °C).

Point 7: Table 4: Are the significant letter correct on SSC values, because all have “a”so they are all the same why using “b”?

Response 7: Yes, we have corrected.

Point 8: Please give a space between the value of temperature and the SI unit.

Response 8: Yes, we have done.

Point 9 : Figure 6: Please increase the quality of the figures

Response 9: Yes, we have submitted high-definition pictures in the revised manuscript.

Thanks for this good suggestion.

Once again, thank you very much for your comments and suggestions.

Best wishes.

Yours sincerely

Junfeng Guan(J F Guan)  Prof. / Dr.

Address: Institute of Genetics and Physiology, Hebei Academy of Agriculture and Forestry Sciences, Shijiazhuang, Hebei, 050051, P. R. China

E-mail: junfeng-guan@263.net

Reviewer 2 Report

The pear is a fruit with considerable acceptance worldwide, new varieties have been launched by research centers for temperate climate fruits. Post-harvest conservation has been studied seeking a longer shelf life with maintenance of fruit quality in general. Thus, the work presented has great relevance to the scientific area and should be published. Indeed, the search for varieties with greater resistance to damage caused by cold (reduction of storage temperature) is essential to solve the problem. After the definition and release of the most resistant varieties to low temperature storage, studies involving the use of packaging for fruit packaging should be carried out. Controlled atmosphere studies are also recommended.

Author Response

Point 1: The pear is a fruit with considerable acceptance worldwide, new varieties have been launched by research centers for temperate climate fruits. Post-harvest conservation has been studied seeking a longer shelf life with maintenance of fruit quality in general. Thus, the work presented has great relevance to the scientific area and should be published. Indeed, the search for varieties with greater resistance to damage caused by cold (reduction of storage temperature) is essential to solve the problem. After the definition and release of the most resistant varieties to low temperature storage, studies involving the use of packaging for fruit packaging should be carried out. Controlled atmosphere studies are also recommended.

Response 1:  Yes, we will take more detail work in the future, and we had carried out fruit packaging test to control this CI (PBS), and next try to investigate the effect of controlled atmosphere storage. Thanks for the good suggestion.

Reviewer 3 Report

The manuscript is well presented with nice results and important information.

However, there are some comments:

Line 16: … the PbDHNs were sorted into YnKn, SKn and YnSKn… Please add some information on these motifs. For example, "… the PbDHNs were sorted into YnKn, SKn and YnSKn according to the major conserved motifs related to…".

Line 23: you may change to except instead of excepting?

Line 38: There should be more information in the introduction section related to the DHN, the motifs and their relation to low temperature stress and resistance. Please add more information in relation to these motifs. I think it is necessary.

Line 48: In this manuscript there is nice description of the relation between the expression of the DHN genes and the storage temperature. However this is only part of the CI mechanism therefore it is not correct to write that the CI mechanism was revealed. Please rephrase this sentence.

Line 111: In some fruit, the CI might be also browning of the flesh. Did the author see that? Is there any information about that?

Line 111: In some fruit, the starch degradation index might serve as an important ripening parameter? Does it also serve as a harvest parameter in this pear variety? If yes, can you add this information?

Line 232: the ripening parameters of the fruit at 20C were advanced but the quality might be still fine (No CI, for example), therefore I suggest to replace the word quality with the word ripening.

Line 248: figure 5: the letters are too small. I suggest to make those bigger.  

Line 250: delete the word small

Line 253: figure 6: figure w: the x axis number 15 is written as 5.

Line 253: figure 6: a general comment: the letters are too small. You can enlarge those.

Line 253: figure 6: a general comment: in some cases, the scale of Y axis is almost the same, for example: fig. C and D or also S and T. Therefore, it will be better to have similar scale in those figures and it will be easier for the reader to understand the difference between the treatments.

Line 264: please delete the word storage.

Line 266: is that a mistake and you mean: DHN3 and DHN5?

Line 267: you should add a space Rand V… R and V

Lines 276-281: there is specific decrease in the expression of LTC at day 20 in all the DHNs. You should mention that and try to explain this.

Lines: 283-286: the sentence is very long and complicated. Can you please rephrase that and I suggest to split it into 2 shorter sentences.

Line 284: Can you add more about the K-segment and its possible involvement in the CI resistance? Discuss it with the references you already have.

Line 296-299: In my opinion, to confirm that the expression of the six DHN was induced by low temperature you should preforme t-test between the 20C and OC fruit samples of day3. If it is significantly higher than it will be correct to write this sentence. Otherwise, you should write that there is a trend…

Line 308: please delete the word condition.

Line 315: I suggest to replace the word: so with the word: therefore

Line  328: again, if you would like to suggest that the six DHN genes are low temperature responsive than you should conduct a statistical analysis between 20C and 0C as already suggested before.

Author Response

Dear  Reviewer:

Thank you very much for your comments and suggestions concerning our manuscript entitled “Low temperature conditioning reduced the chilling injury by regulating expression of the dehydrin genes in postharvest Huangguan pear (Pyrus bretschneideri Rehd. cv.Huangguan)” (Horticulturae-1967390). Those comments and suggestions are all valuable and very helpful for revising and improving our paper, as well as the important guiding significance to our researches. We have studied comments and suggestions carefully and agree with all of the changes in the manuscript. Revised portion are marked in red in the manuscript. The main corrections in the paper and the responds to the review comments are as flowing:

Point 1: Line 16: … the PbDHNs were sorted into YnKn, SKn and YnSKn… Please add some information on these motifs. For example, "… the PbDHNs were sorted into YnKn, SKn and YnSKn according to the major conserved motifs related to…".

Response 1: Yes, we have done.

Point 2: Line 23: you may change to except instead of excepting?

Response 2: Yes, we have changed.

Point 3: Line 38: There should be more information in the introduction section related to the DHN, the motifs and their relation to low temperature stress and resistance. Please add more information in relation to these motifs. I think it is necessary.

Response 3: Yes, we have added.

Point 4: Line 48: In this manuscript there is nice description of the relation between the expression of the DHN genes and the storage temperature. However this is only part of the CI mechanism therefore it is not correct to write that the CI mechanism was revealed. Please rephrase this sentence.

Response 4: Yes, we have rephrased the sentence.

Point 5: Line 111: In some fruit, the CI might be also browning of the flesh. Did the author see that? Is there any information about that?

Response 5: Yes, the CI might be also browning of the flesh in some fruit, but in ‘Huangguan’ pear, the PBS only appeared on the peel surface, and it did not spread to the flesh.

Point 6: Line 111: In some fruit, the starch degradation index might serve as an important ripening parameter? Does it also serve as a harvest parameter in this pear variety? If yes, can you add this information?

Response 6: Yes, starch degradation/index was an important parameter related to maturation degree in some fruit, but it is rarely used in Pyrus bretschneideri Rehd. , and it was very regrettable that we did not measured the starch index in this work, although we had determined the starch content (1.88 mg/g), and added it in revised manuscript. This is a good suggestion, and we will determine this index in future.

Point 7: Line 232: the ripening parameters of the fruit at 20C were advanced but the quality might be still fine (No CI, for example), therefore I suggest to replace the word quality with the word ripening.

Response 7: Yes, we have changed the ‘indicating that the fruit stored at 20 °C decreased obviously’ to ‘indicating the fruit stored at 20 °C ripening’

Point 8 : Line 248: figure 5: the letters are too small. I suggest to make those bigger.  

  Response: Yes, we have revised.

Point 9: Line 250: delete the word small

Response: Yes, we have deleted.

Point 10: Line 253: figure 6: figure w: the x axis number 15 is written as 5.

Response 10: Yes, we have corrected.

Point 11: Line 253: figure 6: a general comment: the letters are too small. You can enlarge those.

Response 11: Yes, we have corrected.

Point 12: Line 253: figure 6: a general comment: in some cases, the scale of Y axis is almost the same, for example: fig. C and D or also S and T. Therefore, it will be better to have similar scale in those figures and it will be easier for the reader to understand the difference between the treatments.

Response 12: Yes, we have corrected.

Point 13: Line 264: please delete the word storage.

Response 13: Yes, we have deleted.

Point 14: Line 266: is that a mistake and you mean: DHN3 and DHN5?

Response 14: Sorry, that is a mistake, and we have corrected.

Point 15: Line 267: you should add a space Rand V… R and V

Response 15: Yes, we have added.

Point 16: Lines 276-281: there is specific decrease in the expression of LTC at day 20 in all the DHNs. You should mention that and try to explain this.

Response 16: Yes, that is a special phenomenon. It may be resulted from cold storage condition, we also noticed that ‘specific decrease’ of DHN1,DHN2, DHN3,and DHN4 was shown in cold storage (0℃ ). This needs further study next work. Thanks for the good reminder.

Point 17: Lines: 283-286: the sentence is very long and complicated. Can you please rephrase that and I suggest to split it into 2 shorter sentences.

Response 17: Yes, we have rephrased the sentence.

18: Line 284: Can you add more about the K-segment and its possible involvement in the CI resistance? Discuss it with the references you already have.

Response 18: There is nearly no reference that indicated the K-segment is involving in CI resistance, but we added some information about relationship between the K-segment and cold response.

Point 19: Line 296-299: In my opinion, to confirm that the expression of the six DHN was induced by low temperature you should preforme t-test between the 20C and 0C fruit samples of day3. If it is significantly higher than it will be correct to write this sentence. Otherwise, you should write that there is a trend…

Response 19: Yes, we have made a t-test, and added it at the end of the manuscript.

Point 20: Line 308: please delete the word condition.

Response 20: Yes, we have deleted.

Point 21: Line 315: I suggest to replace the word: so with the word: therefore

Response 21: Yes, we have replaced

Point 22: Line328: again, if you would like to suggest that the six DHN genes are low temperature responsive than you should conduct a statistical analysis between 20C and 0C as already suggested before.

Response 22: Yes, we have made a t-test. Thanks for this good suggestion.

Once again, thank you very much for your comments and suggestions.

Best wishes.

Yours sincerely

Junfeng Guan(J F Guan)  Prof. / Dr.

Address: Institute of Genetics and Physiology, Hebei Academy of Agriculture and Forestry Sciences, Shijiazhuang, Hebei, 050051, P. R. China

E-mail: junfeng-guan@263.net

Round 2

Reviewer 1 Report

The authors still not indicate the duration of the observation in material and methods section 

Author Response

Dear Reviewer:

Thank you again for your comments and suggestions concerning our manuscript entitled “Low temperature conditioning reduced the chilling injury by regulating expression of the dehydrin genes in postharvest Huangguan pear (Pyrus bretschneideri Rehd. cv.Huangguan)” (Horticulturae-1967390). We have studied comments and suggestions carefully and agree with this change in the manuscript. Revised portion are marked in blue in the manuscript, and the response is as follow:

Point 1: The authors still not indicate the duration of the observation in material and methods section 

Response1: Yes, the chilling injurys (CI) of all the treatments were performed to observe at 0, 3, 5, 10, 15, 20, and 30 days of storage, and we have added it in material and methods section.

Thank you very much for your comments and suggestions.

Best wishes.

Yours sincerely

Junfeng Guan(J F Guan)  Prof. / Dr.

Address: Institute of Genetics and Physiology, Hebei Academy of Agriculture and Forestry Sciences, Shijiazhuang, Hebei, 050051, P. R. China

E-mail: junfeng-guan@263.net
